

# Deficiencies of effectiveness of intervention studies in veterinary medicine: a cross-sectional survey of ten leading veterinary and medical journals

Nicola Di Girolamo[1,2,3] and Reint Meursinge Reynders[4,5]

[1] Department of Veterinary Sciences, University of Bologna, Bologna, Italy
[2] Centro Veterinario Specialistico, Roma, Italy
[3] EBMVet, Cremona, Italy
[4] Department of Oral and Maxillofacial Surgery, Academic Medical Center, University of Amsterdam, Amsterdam, The Netherlands
[5] Private practice of orthodontics, Milan, Italy

Corresponding author
Nicola Di Girolamo,
nicoladiggi@gmail.com

## ABSTRACT

The validity of studies that assess the effectiveness of an intervention (EoI) depends on variables such as the type of study design, the quality of their methodology, and the participants enrolled. Five leading veterinary journals and 5 leading human medical journals were hand-searched for EoI studies for the year 2013. We assessed (1) the prevalence of randomized controlled trials (RCTs) among EoI studies, (2) the type of participants enrolled, and (3) the methodological quality of the selected studies. Of 1707 eligible articles, 590 were EoI articles and 435 RCTs. Random allocation to the intervention was performed in 52% (114/219; 95%CI:45.2–58.8%) of veterinary EoI articles, against 87% (321/371; 82.5–89.7%) of human EoI articles (adjusted OR:9.2; 3.4–24.8). Veterinary RCTs were smaller (median: 26 animals versus 465 humans) and less likely to enroll real patients, compared with human RCTs (OR:331; 45–2441). Only 2% of the veterinary RCTs, versus 77% of the human RCTs, reported power calculations, primary outcomes, random sequence generation, allocation concealment and estimation methods. Currently, internal and external validity of veterinary EoI studies is limited compared to human medical ones. To address these issues, veterinary interventional research needs to improve its methodology, increase the number of published RCTs and enroll real clinical patients.

## INTRODUCTION

Evaluation of medical interventions may be biased by inferential reasoning. The trustworthiness and applicability of outcomes of studies that assess the effectiveness of an intervention (EoI) strongly depend on the (1) choice of the research design, (2) the methodological quality, and (3) the characteristics of the included population (*Schulz et al., 1995*; *Moinpour et al., 2000*; *Moher et al., 2010*). These issues apply to both veterinary

and human medicine. In this study we assessed how these three items differ between these fields.

A randomized controlled trial (RCT) is a study in which patients are randomly allocated to either an intervention or a control group (*Sackett et al., 1996*; *Sibbald & Roland, 1998*). This design avoids most of the biases that occur in observational studies and has the potential to provide the highest quality of evidence (*Moher et al., 2010*). Ideally, the effectiveness of every intervention should be tested in a RCT before its implementation in clinical practice (*Moher et al., 2010*; *Armitage, 1982*). Although the importance of RCTs is universally acknowledged, the prevalence of RCTs varies between different specialties. This prevalence could be used to assess how robust is research on EoI in a specific field. For example, in periodontal research 10% of the publications between 1980 and 2000 were RCTs (*Sjögren & Halling, 2002*). In nursing science, 3.8% of all articles published between 1986 and 2000 were RCTs (*Jiang et al., 2002*). In plastic surgery journals, RCTs comprised only 1.8% of all published articles between 1990 and 2005 (*Momeni et al., 2008*). The prevalence of RCTs was not constant over the last 50 years (*Sjögren & Halling, 2002*; *Becker et al., 2008*). In a survey including all specialties of internal medicine, a common ascending trend in the publication of RCTs was found between 1966 and 2001 (*Strippoli, Craig & Schena, 2004*), while between 1998 and 2002 there was no increase. Similarly, in periodontal research there was an increase in the annual number of RCTs published between 1980 and 1994, but the number remained approximately unchanged between 1994 and 2000 (*Sjögren & Halling, 2002*). Studies that assessed the prevalence of RCTs among EoI studies were not identified in the published veterinary literature.

The label "randomized" is not sufficient to guarantee the methodological soundness of an EoI study. In fact, RCTs should also adhere to a wide variety of other quality parameters (*Schulz et al., 1995*; *Moher et al., 2010*). When the RCTs published in the veterinary literature were evaluated, several methodological issues were identified (*Elbers & Schukken, 1995*; *Lund, James & Neaton, 1998*; *Brown, 2006*; *Brown, 2007*; *Sargeant et al., 2009*; *Sargeant et al., 2010*; *Giuffrida, Agnello & Brown, 2012*; *Giuffrida, 2014*). *Lund, James & Neaton (1998)* identified a lack of reporting of at least 2 of 6 evaluated domains (random sequence generation, informed consent, eligibility criteria, blinding, power calculation and statistical analyses) in all RCTs on dogs and cats published between 1986 and 1990. *Brown (2006)* found that only 11% of 97 RCTs published between 2000 and 2005 on dogs and cats reported both random sequence generation and allocation concealment. In a subset of 63 RCTs on dogs and cats, *Brown (2007)* reported that most RCTs with losses to follow-up did not account for these losses in the data analysis and did not acknowledge the potential impact on the outcomes of these RCTs. Another study found substantive deficiencies in the reporting of key methodological domains in RCTs with dogs and cats published between 2006 and 2008 (*Sargeant et al., 2010*). RCTs performed in laboratory animal research showed similar methodological problems: an overview of 31 systematic reviews found that only 29% of studies reported randomization, 15% of studies reported allocation concealment, and 35% of studies reported blinded outcome assessment (*Hirst et al., 2014*).

Even when RCTs are conducted according to high methodological standards (i.e., strong internal validity), their applicability can be challenged by the type of patients enrolled. Outcomes of RCTs have limited external validity when the participants of the study represent only a small part of the population of interest (*Pressler & Kaizar, 2013*), e.g., if participants are healthier than patients that are not recruited (*Halbert et al., 1999*; *Moinpour et al., 2000*). This issue can affect the external validity of research outcomes and limits the translation of results from RCTs to "true" clinical patients.

The veterinary literature shares several common factors with the medical literature (e.g., presence of generalist and specialist journals). Furthermore, opinion leaders have recently underscored that animal trials should be more similar to human RCTs (*Muhlhausler, Bloomfield & Gillman, 2013*). We therefore decided to exploit the similarities between these two disciplines, and to use human medical literature as the comparator for the veterinary medical literature.

The purpose of this study is to compare the veterinary literature and human medical literature regarding (1) the prevalence of RCTs among EoI studies; (2) the prevalence of RCTs that did not enroll clinical patients; (3) the reporting of key methodological domains in RCTs. Prior to undertaking this research study we conducted scoping searches of the literature to avoid research replication (*Chalmers et al., 2014*). These searches showed that no previous cross-sectional literature reviews compared the quality of EoI articles in veterinary medicine with those in human medicine.

## MATERIALS AND METHODS

### Design and study outcomes

We performed a cross-sectional analysis of the literature comparing leading veterinary medical and human medical journals in the year 2013.

Primary outcomes for this study were (1) the difference in the prevalence of RCTs between veterinary and human medical literature; (2) the difference in the prevalence of enrollment of participants that are not real clinical patients between veterinary and human medical literature; (3) the difference in reporting of key methodological domains in RCTs between veterinary and human medical literature. All other outcomes are considered secondary outcomes.

### Sample size and included journals

We performed a pilot study to assess the prevalence of RCTs over the total number of articles that described the effectiveness of interventions in veterinary medicine and human medicine. All articles that were published in the first six months of 2006 in one veterinary journal (JAVMA) and one human medical journal (JAMA) were assessed for the prevalence of RCTs. A prevalence of 69.1% (29 RCTs/42 intervention articles) and 29.7% (8 RCTs/27 intervention articles) was identified for JAMA and JAVMA respectively. Using a formula for two proportions and equal group size (*Kirkewood & Sterne, 2003*) we calculated that a minimal sample of 45 articles per group was required to have 90% power to detect a difference at a level of statistical significance of 1%. We estimated how many journals we had to search and for which time span based on pertinent research data from a

study by *Giuffrida, Agnello & Brown (2012)*. Based on the pilot study we expected to find roughly one RCT per 3 EoI articles and that each journal should have published approximately 15 EoI articles per-year. To be conservative, we included 5 leading journals for each discipline. The "VETERINARY SCIENCES" and "MEDICINE, GENERAL AND INTERNAL" categories of the 2013 ISI Journal Citation Report were sorted by impact factor. All the journals that focused on sub-specialties or in non-English language were excluded. All the journals that were not published before the year 2000 or had an impact factor lower than 1.0 were excluded. Aims and scopes of the remaining journals were evaluated on their websites until the first 5 journals presenting broad scope were identified. Veterinary journals included were: '*Veterinary Journal*' (Vet J); '*Veterinary Record*' (Vet Rec); '*Journal of Veterinary Internal Medicine*' (JVIM); '*Journal of the American Veterinary Medical Association*' (JAVMA); '*American Journal of Veterinary Research*' (AJVR); Medical journals: '*New England Journal of Medicine*' (NEJM); '*the Lancet*'; '*Journal of the American Medical Association*' (JAMA); '*British Medical Journal*' (BMJ); '*Annals of Internal Medicine*'. Impact factors ranged from 1.2 to 2.2 for veterinary medical journals and from 16.1 to 54.4 for human medical journals. Details of the pilot study, sample size calculation and eligibility criteria for journal inclusion are reported in Supplementary Note 1.

## Data extraction

We hand-searched all full-text articles of all the issues of the 10 selected journals published between the 1st of January 2013 up to the 31st of December of 2013, including supplements, through their online archives. The total number of full original articles, EoI articles, and RCTs were recorded. EoI articles were subsequently classified based on their inclusion of real patients and type of interventions (surgical/non-surgical). RCTs were further classified based on their methodological characteristics.

### *Classification of the EoI articles*

We extracted the following data items:

*Number of full original articles:* primary research, including subgroup analysis or follow-ups of previous articles; case series, defined as original reports including more than one patient.

*Number of articles evaluating effectiveness of interventions (EoI):* "*Effectiveness*" was defined as "*evaluation of benefits*" of an intervention. "*Interventions*" were defined as "*acts used to improve health, to treat a particular condition or disease in process or to prevent development of a particular condition or disease*" (*Farlex Medical Dictionary, 2014*; *Merriam-Webster Encyclopaedia Britannica, 2014*). Studies eligible as "EoI articles" included: Case series, case-control studies, cohort studies, analytical cross-sectional studies, non-randomized controlled trials, and RCTs.

*Number of EoI articles that described surgical interventions:* Studies of surgical interventions face different challenges regarding several aspects, including study design (*McCulloch, 2009*). To account for this factor, the type of intervention was categorized

as surgical/non-surgical. EoI articles were considered "surgical" when (1) the intervention required cutting of the skin. Needle-related procedures (e.g., amniocentesis, etc.) were not considered surgical procedures; (2) The difference between the control and the experimental group was based on the type or technique of the surgical procedure. A trial was not considered "surgical" when the difference between the control and the experimental group was a medication given either before or after a surgical procedure.

*Number of RCTs:* Studies were defined RCTs based on the US National Library of Medicine 2008 definitions for the Publication Type terms 'Randomized Controlled Trial' and based on the definition of the Cochrane glossary (*Cochrane Community, 2014*). All the reports with allocation to interventions described as "randomized" were included (*Schulz et al., 1994*). A study was classified as "a RCT" when (1) at least two interventions were compared; (2) and randomization was mentioned.

*Number of RCTs that included real patients:* We evaluated if RCTs involved real clinical patients or non-patients. Real clinical patients were defined as "*the population that presents the condition that needs to be treated or prevented and that will benefit of the intervention once established.*" Non-patients refer to voluntary healthy participants or laboratory animals. Articles were considered to include real clinical patients when these individuals or animals: (1) suffered from a spontaneous disease; and (2) were exposed to real-life conditions.

The full definitions of each extracted data item are reported in Supplementary Note 2. Categorization of the EoI articles in randomized and non-randomized trials was initially performed through scrutinizing the title and the abstract of the article. If random allocation was not mentioned in the title and the abstract, the full text was searched for the term "random." Full-texts of all the articles classified as "RCTs" were retrieved for the second phase of the study.

### Evaluation of the RCTs

We assessed all the RCTs retrieved from the 5 selected veterinary journals in 2013 for the reporting of key methodological items (*Chan & Altman, 2005*). Random sampling stratified by journal was performed on the medical RCTs in order to assess a number of medical RCTs in a ratio of 1:2 compared with veterinary RCTs. Since 114 veterinary RCTs were retrieved, a total of 60 medical RCTs were sampled. Each medical RCT was sequentially numbered and 5 series of 12 random numbers (one for each journal) were generated with a random number generator.

Two operators independently assessed the electronic full-text of each RCT and eventual supporting information. In case of disagreement, an arbiter was consulted. The following key methodological domains were assessed: primary outcome, power calculation, random sequence generation, allocation concealment, blinding of participants, blinding of personnel, blinding of outcome assessors, intention-to-treat, effect size estimation methods (Table 1). Details regarding research procedures and definitions are reported in Supplementary Note 2.

**Table 1 Definitions used to assess characteristics of publications from 5 leading veterinary and 5 leading medical journals in 2013.** Reporting of methodological domains was assessed in all the randomized controlled trials (RCTs) extracted. Full definition of each item is given in the main text and supplementary files.

| | Terms | Descriptions |
|---|---|---|
| **Characteristics of articles** | *Original articles* | Primary research, including subgroup analyses, follow-ups of previous article and case series |
| | *Effectiveness of intervention (EoI) articles* | Primary research evaluating the benefits of an intervention |
| | *Randomized controlled trials (RCT)* | EoI studies with allocation to interventions reported as randomized |
| | *Real patients RCTs* | RCTs that included individuals or animals that suffered from a spontaneous disease and were exposed to real-life conditions |
| | *Surgical (RCT/EoI) articles* | Same as previous definitions, but evaluating the benefits of a surgical intervention |
| | *Explicit RCT* | Trials registered in a trial repository or self-defining "randomized controlled trial" |
| | *Explicit parallel RCT* | Same as pervious, but employing only two arms |
| | *Standalone RCT* | Lack of additional non-randomized work (i.e., in vitro or prospective data) reported in the same article of the RCT |
| | *Crossover RCT* | RCT in which participants receive a sequence of different treatments |
| | *Cluster RCT* | RCT in which groups of participants are randomized to different treatments |
| **Key methodological domains evaluated in RCTs** | *Primary outcome* | A primary outcome is explicitly reported in the published article |
| | *Power calculation* | A power calculation performed a priori to estimate the sample size is explicitly reported |
| | *Random sequence generation* | Methods employed to generate the random list and type of randomization are explicitly reported |
| | *Allocation concealment* | Methods used to prevent the individuals enrolling trial participants from knowing or predicting the allocation sequence in advance are explicitly described in the article |
| | *Blinding of participants* | Explicit description that participants/pet owners were unaware of participants' group allocation |
| | *Blinding of personnel* | Explicit description that operators involved in the care of participants were unaware of participants' group of allocation |
| | *Blinding of outcome assessors* | Explicit description that outcome assessors were unaware of participants' group of allocation |
| | *Intention-to-treat* | Explicit mention that the analysis was made on an "intention-to-treat" basis. |
| | *Effect size estimation methods* | Results are reported with methods that estimate the effect size with confidence interval. |

**Note:**
EoI, Effectiveness of intervention.

## Statistical analysis

SPSS statistics (v22.0, IBM, Chicago, IL) was used for the statistical analysis. Statistical significance was set at $P < 0.05$, unless otherwise specified. Prevalence of RCTs in veterinary and human medical journals was calculated as: number of RCTs/total number of EoI articles. Confidence intervals for proportions were estimated with the Wilson procedure with continuity correction (*Newcombe, 1998*). Articles were the unit of the primary analysis. In the unadjusted analysis the strength of the association was described as odds ratios (ORs) with their 95% confidence interval and was calculated in two-by-two tables. Chi-squared tests or Fisher tests (depending on the number of expected values in

each cell) were used to evaluate whether there were significant differences in proportions. If one or more of the cells in the contingency table were zero, a non-constant continuity correction was employed to account for the imbalance of the group sizes. A factor of the reciprocal of the size of the opposite treatment group was added to the cells (*Sweeting, Sutton & Lambert, 2004*).

Logistic regression and multilevel logistic regression models were developed to provide odds ratio adjusted for confounders (*Peng & So, 2002*; *Heck, Thomas & Tabata, 2013*). In the multilevel logistic regression with prevalence of RCT as the outcome, discipline (veterinary/general medicine) and type of intervention (i.e., surgical/non-surgical) were included as fixed effects and journal as a random effect (*Heck, Thomas & Tabata, 2013*). In the logistic regression models, confounders included: type of intervention (i.e., surgical/non-surgical), type of trial (randomized/non- randomized), type of patients enrolled (clinical patients/non-patients), and discipline (veterinary/medicine). Variables were retained in the model based on statistical significance (P < 0.1) and based on the effect they had on the final model. To avoid overfitting of the model, a minimum of 10 events per predictor variable were required (*Peduzzi et al., 1996*). Goodness of fit was assessed with Hosmer-Lemshow test. Nagelkerke R squared was reported (*Nagelkerke, 1991*). Multicollinearity was suspected with variance inflation factor >3 and condition index >30 (*Midi, Sarkar & Rana, 2010*).

To identify differences in the number of patients enrolled, the non-parametric Mann-Whitney U test was used as indicated by the distribution (*Lehmann, 1951*).

Sensitivity and subgroup analyses planned a priori were conducted to support the robustness and the generalizability of the association between a specific discipline and the quality of reporting of key methodological domains. Sensitivity analyses were conducted by excluding from the analysis certain types of randomized trials, in order to determine their effect on the final results. In the first sensitivity analysis, we excluded all the RCTs that did not enroll real clinical patients because an earlier study identified a lack of reporting of key methodological items in RCTs on laboratory animals (*Hirst et al., 2014*). In an additional sensitivity analysis we excluded all the surgical RCTs because previous research on such RCTs has found a lack of adequate reporting of key methodological items (*Sinha et al., 2009*). In a further analysis, we included only "explicit RCTs" (i.e., randomized trials that were explicitly defined in the text as "randomized controlled trial" or trials registered in a repository) because we hypothesized that such RCTs would have been conducted with a special focus on high methodological standards. In the final analyses, we excluded all the cross-over RCTs and cluster RCTs because these trial designs have also been associated with poor reporting (*Walleser, Hill & Bero, 2011*; *Straube, Werny & Friede, 2015*). Results of sensitivity analyses were reported with forest plots generated with RevMan (5.3; Copenhagen: The Nordic Cochrane Centre, The Cochrane Collaboration, 2014).

## RESULTS

### Effectiveness of intervention articles

A total of 1707 eligible articles were identified through hand-searching procedures in 10 selected journals for the year 2013 (Fig. 1). Of these articles, 990 were published in

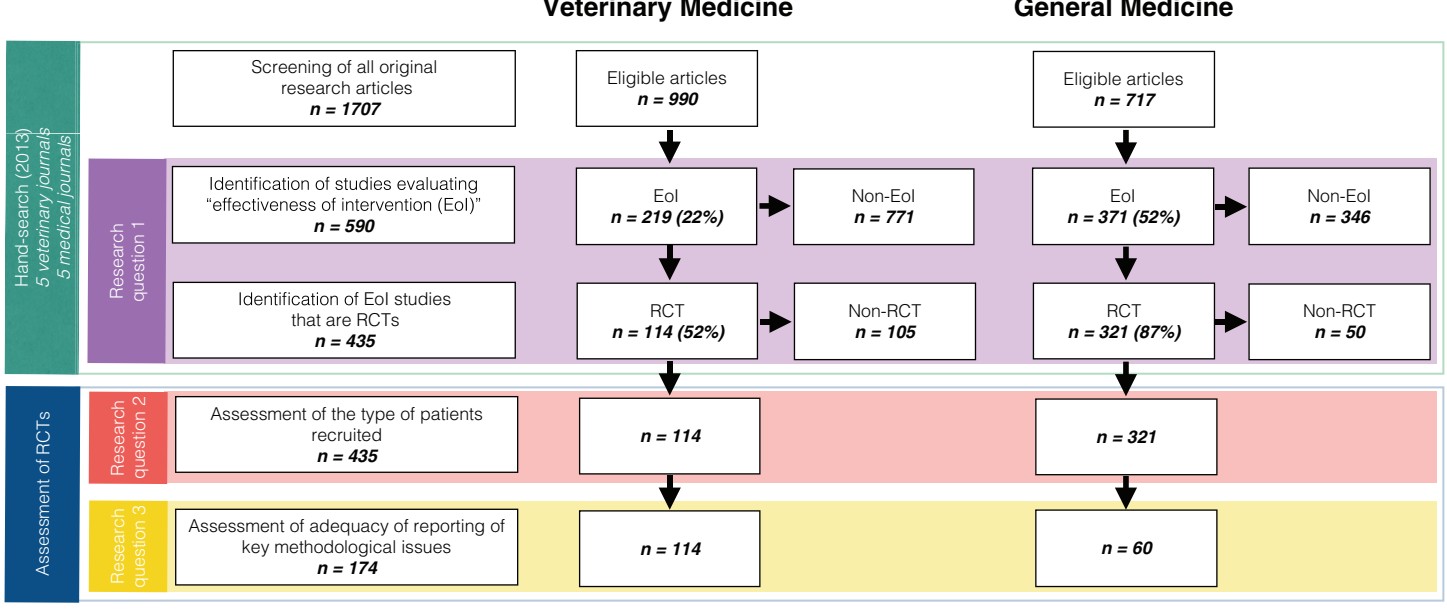

**Figure 1  Study flow.**

veterinary medical journals, and 717 in human medical journals. The distribution of eligible articles among the specific journals is depicted in Table 2. Overall, 35% of the eligible articles (590/1707) were EoI studies. Effectiveness of an intervention was evaluated in 22% (219/990) of the veterinary medical articles and in 52% (371/717) of the human medical articles.

## Randomized controlled trials
### Prevalence of RCTs
The prevalence of RCTs among the EoI articles was 52% (114/219; 95% CI: 45.2% to 58.8%) in veterinary medical journals, versus 87% (321/371; 95% CI: 82.5% to 89.7%) in human medical journals. The prevalence of RCTs (number of RCTs/number of EoI articles) in the various journals is reported in Table 2. EoI articles published in veterinary medical journals had 5.9 times the odds of being non-randomized, compared with articles in human medical journals (OR: 5.9; 95% CI: 4 to 8.8; P < 0.001). In the adjusted analysis, including "type of intervention" (surgical/non-surgical) as fixed effect and "journal" as random effect, the results were substantially unchanged (OR: 5.6; 95% CI: 3.1 to 9.8). Human medical journals had 6 to 15 times the odds of publishing a RCT compared with one of the veterinary medical journals (Supplementary Table S1). When only "explicit RCTs" were considered, the prevalence of RCTs declined to 21% in veterinary medicine (47/219; 95% CI: 16.3% to 27.6%), while the prevalence remained unchanged (87%) in human medicine.

### Characteristics of RCTs
The number of subjects randomized in the articles ranged from 5 to 28244, with a median of 59 subjects (IQR: 307 subjects), a mean of 857 subjects, and a SD of 3278 subjects.

**Table 2** Number of eligible articles, EoI articles and prevalence of RCTs in 10 leading veterinary and medical journals in 2013.

| | Veterinary Journals | | | | | | Medical Journals | | | | | |
|---|---|---|---|---|---|---|---|---|---|---|---|---|
| | AJVR | JAVMA | JVIM | Vet J | Vet Rec | Total | Annals | BMJ | JAMA | Lancet | NEJM | Total |
| Eligible articles | 193 | 188 | 156 | 320 | 133 | 990 | 75 | 128 | 157 | 152 | 205 | 717 |
| EoI articles | 43 | 51 | 40 | 59 | 26 | 219 | 31 | 46 | 71 | 94 | 129 | 371 |
| RCT (n) | 34 | 18 | 19 | 30 | 13 | 114 | 24 | 37 | 61 | 84 | 115 | 321 |
| (%) | 79% | 35% | 47% | 51% | 50% | 52% | 77% | 80% | 86% | 89% | 89% | 87% |
| Non-RCT (n) | 9 | 33 | 21 | 29 | 13 | 105 | 7 | 9 | 10 | 10 | 14 | 50 |
| (%) | 21% | 65% | 53% | 49% | 50% | 48% | 23% | 20% | 14% | 11% | 11% | 13% |

Notes:

AJVR, American Journal of Veterinary Research; JAVMA, Journal of the American Veterinary Medical Association; JVIM, Journal of Veterinary Internal Medicine; Vet J, Veterinary Journal; Vet Rec, Veterinary Record; Annals, Annals of Internal Medicine; BMJ, British Medical Journal; JAMA, Journal of the American Medical Association; NEJM, New England Journal of Medicine.

Sample size was not normally distributed in veterinary and in human medical trials. Trials were significantly smaller in veterinary medical journals having a median sample size of 26 subjects (range ± IQR: 5–28244 ± 47) compared to 465.5 subjects in human medical journals (range ± IQR: 32–27347 ± 1267) (Mann-Whitney U: 607.0; $P < 0.001$). Furthermore, veterinary crossover trials were significantly smaller, i.e., with a median sample size of 8 subjects (range ± IQR: 5–20 ± 5), compared with 36 subjects in other RCT designs (range ± IQR: 6–28244 ± 63) (Mann-Whitney U: 131.5; $P < 0.001$).

Only 1 out of 114 RCTs (0.9%) in the veterinary journals had a cluster design compared with a 13.3% in the human medical journals (8/60). More than one fifth (21.9%; 25/114) of the veterinary RCTs had a crossover design. Of the 60 randomly sampled human medical RCTs, only one (1.7%; 1/60) had a crossover design, specifically a stepped-wedge design, i.e., cluster and crossover design. Further characteristics of RCTs are reported in the Supplementary Data.

### Prevalence of surgical RCTs

Surgical interventions accounted for 5% of the eligible articles (90/1707), for 15% of the EoI articles (90/590) and for 9.6% of the RCTs (42/435). Surgical interventions had 4.2 times the odds of being published as non-randomized articles than as randomized articles (OR: 4.2; 95% CI: 2.6 to 6.7; $P < 0.001$) (Fig. 2).

No statistically significant differences in the prevalence of surgical EoI articles were found between veterinary and human medicine (18%; [40/219] vs 13%; [50/371]; OR: 0.69; 95% CI: 0.44 to 1.09; $P = 0.11$). However, veterinary RCTs had a notably lower prevalence of surgical interventions compared with veterinary EoI studies (3.5%; 4/114), while the prevalence of surgical interventions remained roughly constant among human medical EoI studies and RCTs (11.8%; 38/321). The strength of the association between surgical intervention and lack of randomization was stronger in veterinary medicine. Human medical journals had 3.7 times the odds of publishing RCTs on surgical interventions than veterinary journals (OR: 3.7; 95% CI: 1.3 to 10.6; $P = 0.01$).

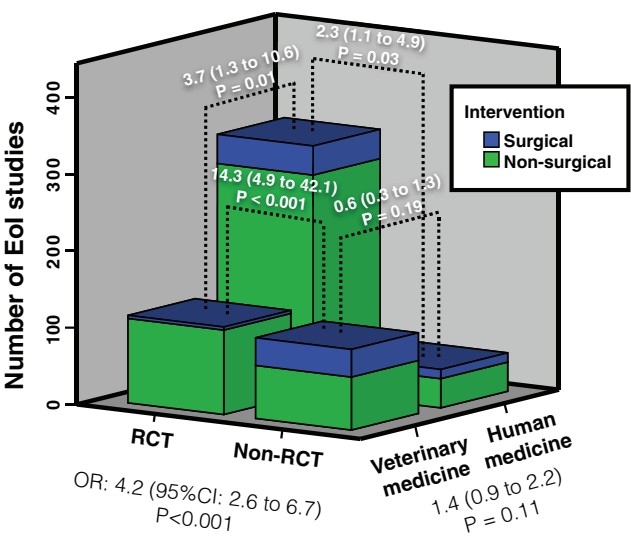

**Figure 2 Association (odds ratio and 95% confidence intervals) between randomization, discipline and type of intervention (surgical/non-surgical).** Notice that the overall prevalence of randomized controlled trials (RCTs) and prevalence of surgical RCTs were lower in veterinary medicine (ORs: 4.2 and 3.7, respectively). However, surgical interventions were more likely to be non-randomized in both disciplines (ORs: 14.3 and 2.3).

### *Enrollment of real clinical patients*

All the human medical RCTs focused on real patients, except one (99.7%; 320/321). Less than a half of the veterinary RCTs (49.1%; 56/114) focused on real patients (OR: 331.4; 95% CI: 45 to 2441.9; P < 0.001).

The vast majority of crossover trials enrolled non-clinical patients (84.0%; 21/25), while 41.6% of the remaining RCTs enrolled non-clinical patients (37/89). Crossover trials had 7.4 times the odds of enrolling non-clinical patients compared with other RCT designs (OR: 7.4; 95% CI: 2.3 to 23.3; P < 0.001).

### Reporting of key methodological domains

The key methodological domains were consistently reported by most human medical RCTs, but irregularly reported by veterinary medical RCTs (Table 3). Blinding procedures of owners, personnel and outcome assessors were the most consistently reported domains in veterinary RCTs. None (0/114) of the veterinary RCTs adequately reported all key methodological domains, while 23% (14/60) of the human medical RCTs reported all those methodological domains (OR: 70.5; 95% CI: 4.1 to 1208.4; P < 0.001). Only 2% (2/114) of the veterinary RCTs, versus 77% (46/60) of the human medical RCTs, adequately reported key methodological domains that are always feasible (OR: 184; 95% CI: 40.2 to 842; P < 0.001). The number of key methodological domains adequately reported was positively correlated with the number of patients enrolled (Spearman r: 0.73; P < 0.001; Fig. 3).

A binary logistic regression model including "reporting of primary outcome," "reporting of allocation concealment," "reporting of random sequence generation,"

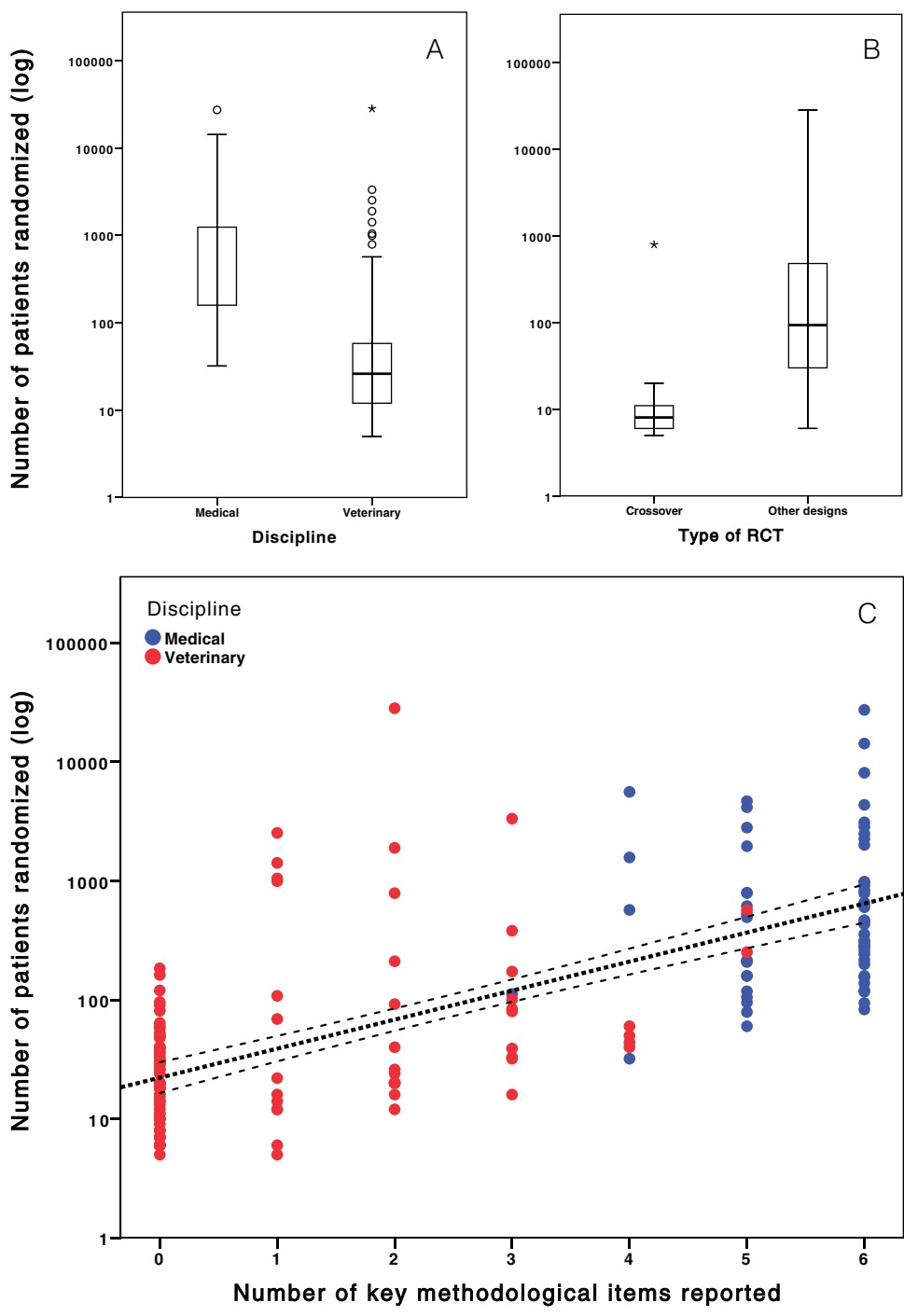

**Figure 3 Number of patients randomized in RCTs and methodological quality.** (A) Difference in number of patients randomized in veterinary and general medicine. (B) Difference in number of patients randomized in cross-over and other study designs. (C) Correlation between the number of methodological issues (primary outcome, power calculation, random sequence generation, allocation concealment, estimation methods, intention-to-treat) reported and number of patients enrolled in each RCT. Notice that the y-axis was plotted on a logarithmic function.

**Table 3 Reporting of key methodological items in RCTs published in leading veterinary and human medical journals.**

| Methodological items | Reporting | Veterinary count | % | Medical count | % | OR | 95% CI | P value |
|---|---|---|---|---|---|---|---|---|
| Power calculation | Stated | 19 | 16.7% | 59 | 98.3% | | | |
| | Not stated | 95 | 83.3% | 1 | 1.7% | 295 | 38.5 to 2261.8 | 0.001 |
| Primary outcome | Defined | 22 | 19.3% | 59 | 98.3% | | | |
| | Not defined | 92 | 80.7% | 1 | 1.7% | 246.7 | 32.4 to 1879.5 | 0.001 |
| Method of random sequence generation | Reported | 23 | 20.2% | 56 | 93.3% | | | |
| | Not reported | 91 | 79.8% | 4 | 6.7% | 55.4 | 18.2 to 168.5 | 0.001 |
| Method of allocation concealment | Reported | 12 | 10.5% | 50 | 83.3% | | | |
| | Not reported | 102 | 89.5% | 10 | 16.7% | 42.5 | 17.2 to 105.0 | 0.001 |
| Detailed blinding of patients/pet owners | Reported | 16 | 14.0% | 30 | 50.0% | | | |
| | Not reported | 98 | 86.0% | 30 | 50.0% | 6.1 | 2.9 to 12.7 | 0.001 |
| Detailed blinding of personnel | Reported | 33 | 28.9% | 21 | 35.0% | | | |
| | Not reported | 81 | 71.1% | 39 | 65.0% | 1.3 | 0.6 to 2.5 | 0.4 |
| Detailed blinding of outcome assessors | Reported | 57 | 50.0% | 44 | 73.3% | | | |
| | Not reported | 57 | 50.0% | 16 | 26.7% | 2.7 | 1.4 to 5.4 | 0.003 |
| Intention-to-treat analysis | Mentioned | 3 | 2.6% | 49 | 81.7% | | | |
| | Not mentioned | 111 | 97.4% | 11 | 18.3% | 164.8 | 44.0 to 617.0 | 0.001 |
| Effect size methods | Used | 18 | 15.8% | 57 | 95.0% | | | |
| | Not used | 96 | 84.2% | 3 | 5.0% | 101.3 | 28.6 to 359.1 | 0.001 |

**Notes:**
OR, Odds ratio; CI, Confidence interval.

"mentioning ITT" and "use of estimation methods" as covariates, was well-fitted (Hosmer and Lemeshow test: P = 0.87) and useful to predict (Nagelkerke R-squared: 0.95) and discriminate (area under the curve: 0.99) the source discipline (i.e., veterinary/medicine) of a RCT. "Reporting of power calculation" was not included in the model because of collinearity between "reporting of the primary outcome" and "reporting of the power calculation." Dependency between the variables after the removal of "reporting of the power calculation" was acceptable for running the logistic regression model (i.e., all condition index lower than 30.0).

### Association between clinical patients and key methodological domains

Veterinary RCTs enrolling non-clinical patients were more likely to lack adequate reporting of each of the methodological domains evaluated in this study (Fig. 4) with the exception of blinding of personnel and intention-to-treat. The lack of an association between reporting and "intention-to-treat" should be interpreted with caution, because only 3 veterinary RCTs reported on this domain.

### Sensitivity and subgroup analyses

Sensitivity analyses supported the robustness of the association between discipline and lack of reporting of key methodological domains (Fig. 5). The amount of surgical RCTs and non-real patient RCTs was not sufficient to perform sensitivity analyses that included these RCTs only.

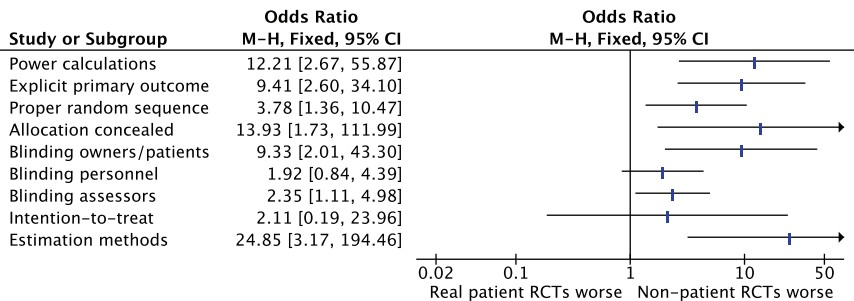

| Study or Subgroup | Odds Ratio M–H, Fixed, 95% CI | Odds Ratio M–H, Fixed, 95% CI |
|---|---|---|
| Power calculations | 12.21 [2.67, 55.87] | |
| Explicit primary outcome | 9.41 [2.60, 34.10] | |
| Proper random sequence | 3.78 [1.36, 10.47] | |
| Allocation concealed | 13.93 [1.73, 111.99] | |
| Blinding owners/patients | 9.33 [2.01, 43.30] | |
| Blinding personnel | 1.92 [0.84, 4.39] | |
| Blinding assessors | 2.35 [1.11, 4.98] | |
| Intention–to–treat | 2.11 [0.19, 23.96] | |
| Estimation methods | 24.85 [3.17, 194.46] | |

**Figure 4 Association between type of patients enrolled (real clinical patients *vs* non-patients) and adequacy of reporting of key methodological issues in RCTs published in leading veterinary journals (n = 114).** M-H, Mantel-Haenszel; CI, Confidence interval.

## DISCUSSION

RCTs, introduced in the early 1940s (*Stuart-Harris, Francis & Stansfeld, 1943*; *Medical Research Council, 1948*), have been widely adopted in clinical research to reduce the risk of subjective evaluation of effectiveness of interventions. Our study showed that the scientific rigor used to evaluate effectiveness of interventions was suboptimal in leading veterinary journals compared with general medical journals. Firstly, we observed a lower prevalence of RCTs in veterinary journals compared with medical journals (research question 1). Secondly, more than half of the RCTs in veterinary medicine did not enroll clinical patients (research question 2). Finally, key methodological domains were underreported in veterinary RCTs compared with those in medical RCTs (research question 3). This latter finding is critical as under-reporting of methodological domains in veterinary RCTs has been associated with increased treatment effects (*Sargeant et al., 2010*).

Observational non-randomized articles are fundamental in certain stages of the development of interventional procedures (*Schünemann et al., 2013*). In veterinary medicine these articles were overabundant and only about half of the articles that assessed the effectiveness of interventions was randomized. A survey in 2006 by *Kuroki, Allsworth & Peipert (2009)* calculated a prevalence of 34.7% (66/190) of RCTs on the total number of articles published in JAMA, Lancet and NEJM. In the present study we observed a prevalence of 50.6% (260/514) of RCTs on the total number of original research articles for the same 3 journals. These data should be considered with caution, because they originated from 2 single articles that applied different sampling strategies. However, there was a 16% increase in the prevalence of RCTs in these journals between 2006 and 2013 (34.7% *vs* 50.6%; Risk difference: 0.16; 95% CI: 0.08 to 0.24). In the present study all journals were hand-searched and all original research articles were scrutinized. Instead, *Kuroki, Allsworth & Peipert (2009)* obtained a random sample of the original research articles. Notwithstanding these differences in selection procedures, the definitions for "journal article" and "RCT" were the same in both studies. If these research studies are considered representative for their specific year of publication, the increase of the prevalence of RCTs in these medical journals in 7 years was impressive. Veterinary journals should also aim to increase the rate of published RCTs.

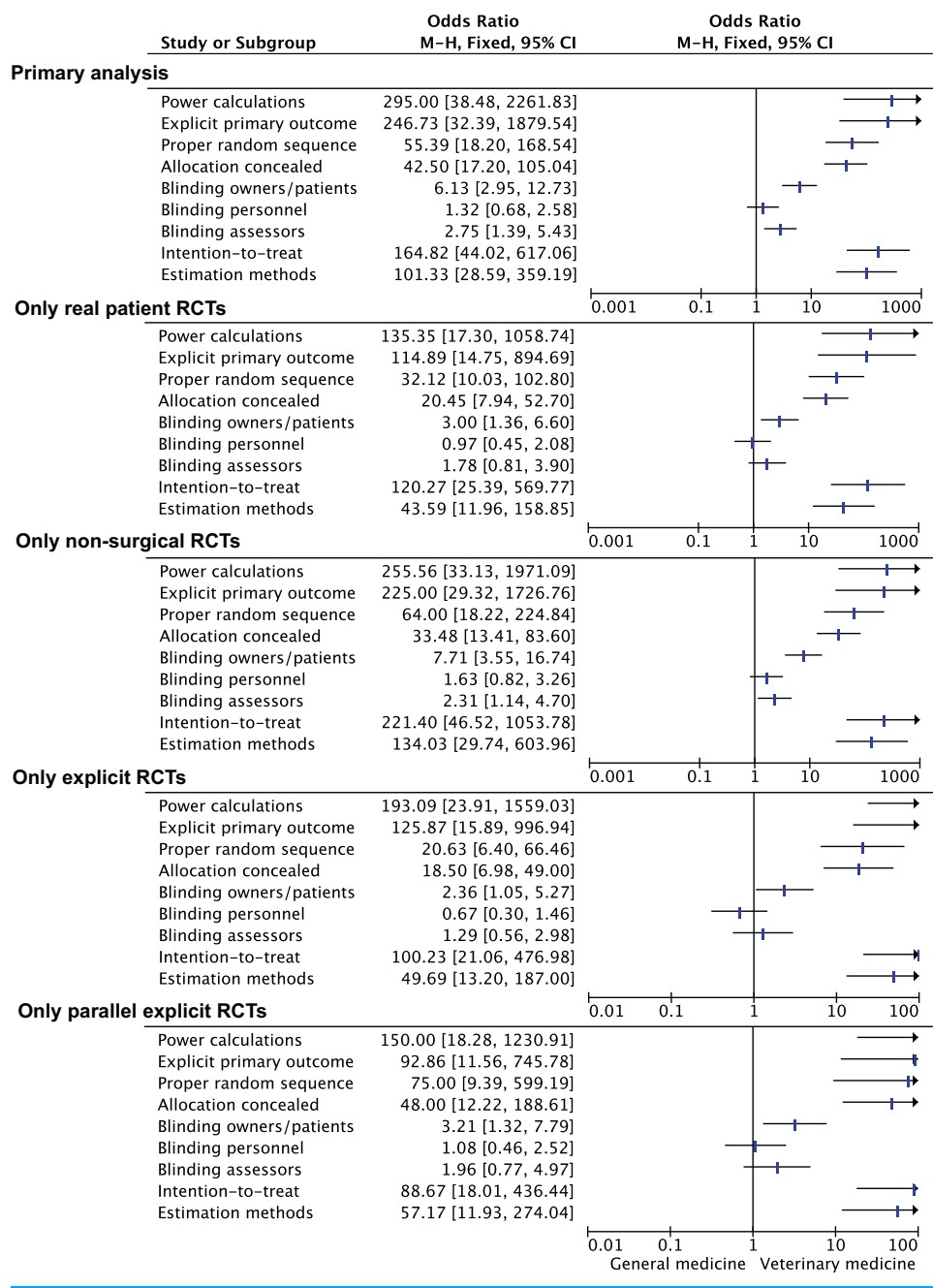

**Figure 5 Sensitivity and subgroup analysis of the association between discipline and reporting of key methodological items.** Primary analysis. Inclusion of explicit RCTs. Exclusion of surgical RCTs. Inclusion of parallel explicit RCTs. Exclusion of non-patient RCTs. M-H, Mantel-Haenszel; CI, Confidence interval.

Experimental animal studies have been recently criticized for being inadequately conducted and reported (*Hirst et al., 2014*; *Sena et al., 2007*; *Landis et al., 2012*). In 2010, the REFLECT statement (Reporting guidelines for randomized controlled trials for livestock and for food safety) and the ARRIVE guidelines (Animal research: reporting of in vivo experiments) were developed for reporting RCTs of respectively livestock and

laboratory animal research (*Kilkenny et al., 2010*; *O'Connor et al., 2010*). However, these guidelines apply only partly to veterinary medicine and often do not cover the needs of veterinary clinicians, i.e., they do not provide guidelines for patient-specific interventions. The finding that just one quarter of the EoI articles in the 5 veterinary journals included is an RCT on clinical patients should be a particular warning sign for veterinary clinicians. Most of the veterinary RCTs are conducted in a pre-clinical setting or in laboratory-controlled conditions and may not apply to the population of interest.

In the present study we observed that crossover trials were rare in leading human medical journals (1 out of 321 RCTs) but frequent in veterinary journals (more than one fifth of the RCTs). Crossover trials in veterinary medicine had a smaller sample size (median sample size of 8 patients) and were more likely to enroll non-clinical patients. The association between non-clinical patients, sample size and crossover design is straightforward in veterinary medicine: the additional expenses related to the use of experimental subjects compared with the use of clinical patients are mainly secondary to purchasing and maintaining them. Therefore, there is a tendency to use the smallest "credible" sample sizes, and to apply crossover designs to provide additional statistical power. The fact that only 1 out of 25 crossover trials presented a power calculation supports this hypothesis. This information is not novel, as *Macleod (2011)* suggested *"In the face of pressures to reduce the number of animals used, investigators often do studies that are too small to detect a significant effect."*

In veterinary journals, parallel explicit RCTs were more likely to report power calculation than crossover trials. However, only 25% of the parallel explicit RCTs reported a power calculation. This is consistent with the results of a study by *Giuffrida (2014)*, who assessed trials of client-owned dogs and cats. The vast majority of medical RCTs reported power calculations, which is consistent with the literature (*Charles et al., 2009*). Veterinary researchers should apply stringent standards of statistical power when planning empirical research.

A statistically significant result does not provide information on the magnitude of the effect and thus does not necessarily mean that the effect is robust (*Landis et al., 2012*). While almost every RCT published in the selected human medical journals provided point estimates with measures of uncertainty (i.e., 95% CI), most articles in veterinary medicine provided just these measures of statistical significance, i.e., 'P values'. Over-reliance on 'P values' when reporting and interpreting results of a RCT may be inappropriate and misleading (*Rothman, 1978*; *Sterne & Davey Smith, 2001*). Veterinary investigators should become more confident with statistical procedures to generate point estimates and measures of uncertainty. Journal editors should encourage authors to employ statistical techniques that maximize the clinical interpretation of RCTs.

Reporting of allocation concealment in RCTs published in human medical journals improved in the last 15 years from 34.4% to 64.7% (*To et al., 2013*). In our study, only one tenth of the RCTs published in veterinary journals mentioned allocation concealment. This is worrying as there is evidence that the lack of allocation concealment is empirically associated with bias (*Schulz et al., 1995*). Researchers should be aware of these consequences and should understand that concealing treatment assignments up to the

point of allocation is always possible regardless of the study topic (*Schulz et al., 1994*). Investigators should therefore implement this issue in veterinary RCTs.

Previous studies evaluating veterinary RCTs focused only on one journal (*Elbers & Schukken, 1995*), others only on some species (*Lund, James & Neaton, 1998*; *Sargeant et al., 2010*) or only on RCTs including client-owned animals (*Brown, 2006*; *Brown, 2007*; *Giuffrida, Agnello & Brown, 2012*). To maximize the external validity of this study, we included all the EoI studies. Nonetheless, the present research study focused on a selected group of leading general journals. In 2006 the proportion of RCTs published among JAMA, Lancet and NEJM was almost 3 times greater than obstetric and gynecology journals (*Kuroki, Allsworth & Peipert, 2009*). In addition, trials published in general medical journals had higher quality scores than those in specialist journals (*To et al., 2013*). Therefore, in this study biases are likely to overestimate the prevalence of RCTs and their methodological quality compared with the remaining journals in human medicine, even if, it cannot be excluded that some specialist journals have a higher prevalence of RCTs due to editorial policies or other factors.

We found that surgical interventions were more likely investigated with non-randomized trials than randomized trials in both disciplines. This is not unexpected, as surgical RCTs pose more challenges than pharmaceutical RCTs (*McCulloch, 2009*). However, the magnitude of this effect may have been amplified by a selection bias. Journals included in this survey were selected for their broad scope. However, it is possible that surgical teams prefer to submit the results of surgical RCTs to specialist surgical journals instead of to broad scope journals. Therefore, future research should include also specialist surgical journals to provide a fairer estimate of the prevalence of these randomized trials.

A possible weakness of this study was the non-blinded assessment of studies, i.e., reviewers knew whether they assessed a veterinary or human medical article. Even though there is some evidence that blind assessments of studies might result in more consistent ratings compared with open assessments (*Jadad et al., 1996*), some other studies suggest that blind assessments provide little benefit (*West, 1997*; *Kjaergard, Villumsen & Gluud, 2001*). A further weakness is the limited sample of included journals. Our selection criteria led to the inclusion of the American Journal of Veterinary Research (AJVR), a journal that tends to focus more on experimental than clinical research. This characteristic could have led to bias toward a lower number of "real patient" RCTs. However, we have performed a sensitivity analysis removing AJVR to evaluate the impact of this single journal on our primary outcomes (prevalence of RCTs, prevalence of real patient RCTs). We observed only a modest change in real patient RCTs (i.e., from 26% to 28%), while there was a further drop in RCT prevalence in veterinary journals (i.e., from 52% to 45%). Therefore, we retain that the impact of this journal on the overall results is limited and it is unlikely that it has significantly biased our conclusions. Another weakness of this study is that we assessed how attrition was reported and dealt with only by means of a proxy (*Chan & Altman, 2005*). This issue is difficult to address, because handling of attrition may be performed in several different ways, and may be reported in different sections of the article (i.e., under statistical analysis but also in the patient flow). As a proxy for this item,

we searched for the wording "intention-to-treat," based on the Cochrane glossary definition (*Cochrane Community, 2014*). Obviously mentioning "intention-to-treat analysis" does not ensure that attrition was properly dealt with, because the term 'intention-to-treat' could have been misused.

A limitation in the generalizability of the study is that we had to obtain a sample of journals due to the number of published journals in the two disciplines (n = 133 in the 'veterinary science' category; n = 154 in the 'medicine, general & internal' category). This sample may not be representative for the entire spectrum of journals in these fields, especially when considering peculiar specialties (e.g., surgery), with exclusive issues in conduct and reporting of RCTs (*McCulloch, 2009*). A larger sample of journals (e.g., inclusion of 10–20 journals per discipline) would have been desirable to obtain results with greater generalizability.

The several limitations we have uncovered in the veterinary literature are probably secondary to the intrinsic difficulties in this field. For example, RCTs are more expensive than observational reports, and veterinary medicine has been historically allotted less funding compared with the human medical field. Furthermore, in veterinary medicine the variety of species makes performing RCTs on every species difficult. The goals in veterinary medicine are also substantially different than in human medicine, e.g. in production medicine, productivity is the primary metric of treatment success.

However, the aforementioned limitations of this study should be considered in the context of the overwhelming magnitude of the identified differences between veterinary and human medical EoI studies. Veterinary researchers should adopt medical research standards by introducing the following pathways for improvement:

1. *Education in basic evidence-based practices at all veterinary levels (i.e., undergraduates to policy-makers).* Education on evidence-based practices effectively engages students (*Aronoff et al., 2010*). Implementing tailored courses for veterinary professionals on conducting, analyzing, and reporting of clinical research is an important starting point.

2. *Implementation of reporting guidelines at all research levels (i.e., researchers, peer-reviewers, journal editors).* Although most veterinary journals adhere to reporting guidelines, such as the CONSORT statement (*Moher et al., 2010*), journals do not require submission of the pertinent checklist together with the manuscript. Implementing this practice would be easy and costless.

3. *Implementation of methodology reviewers in the peer-review process.* Methodology reviewers can provide unique comments during peer-review, which are distinct from the issues raised by regular reviewers (*Day et al., 2002*). As in several human medical journals, veterinary journals should invite methodology reviewers to participate in the peer-reviewing process.

4. *Creation of specific grants for pragmatic randomized controlled trials and predilection of pragmatic randomized controlled trials in grants.* As pragmatic randomized controlled trials provide higher quality evidence than other study designs (*Sibbald & Roland, 1998*), it would be wise to establish purposely-designed grants for pragmatic RCTs, or to favor publication of pragmatic RCTs.

5. *Establishment of a register for veterinary randomized controlled trials.* Trial registration in human medicine was promoted to enhance transparency and accountability in the planning, execution, and reporting of RCTs (*Dickersin & Rennie, 2003*; *De Angelis et al., 2004*). This example should also be adopted in veterinary medicine.

## CONCLUSIONS

The objectives of this study were met, and most of the findings were alarming with regard to both their number and magnitude. We observed that the veterinary literature was characterized by a lower prevalence of randomized articles compared with the human medical literature. More worrying, the vast majority of the RCTs in veterinary medicine lacked adequate reporting of key methodological domains (i.e., primary outcome, power calculation, random sequence generation, allocation concealment, intention-to-treat) and of currently recommended statistical reporting (i.e., effect measures with confidence intervals). Furthermore, RCTs published in veterinary journals were less likely to enroll clinical patients than RCTs published in human medical journals. These outcomes are important, because they could slow down knowledge creation and could result in inappropriate claims of effectiveness of interventions. These findings should be a stimulus for future researchers to improve upon the soundness of their studies. Awareness of the paucity of evidence behind interventions in veterinary medicine is the first step for improvement.

## ACKNOWLEDGEMENTS

The authors acknowledge Paolo Selleri for his critical review and comments and Lucy Nicole Papa Caminiti for assistance with data extraction.

### Funding

The authors received no funding for this work.

### Competing Interests

The authors declare that they have no competing interests. Nicola Di Girolamo is a private practitioner and Reint Meursinge Reynders is self-employed in his private practice of orthodontics.

### Author Contributions

- Nicola Di Girolamo conceived and designed the study, performed the experiments, analyzed the data, wrote the paper, prepared the figures and tables, and reviewed the drafts of the paper.
- Reint Meursinge Reynders performed the experiments and reviewed the drafts of the paper.

## Data Deposition

The raw data will be deposited at Researchgate after a second manuscript has been submitted for peer review.

## Supplemental Information

Supplemental information for this article can be found online at http://dx.doi.org/10.7717/peerj.1649#supplemental-information.

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
