# Peer review of "Deficiencies of effectiveness of intervention studies in veterinary medicine: a cross-sectional survey of ten leading veterinary and medical journals"

_PeerJ, doi:10.7717/peerj.1649_

## Round 0.1 · original submission · Major Revisions

Three reviewers have provided an extensive number of important comments about the manuscript. In your response, you will need to provide detailed responses to each one, and make modifications to your manuscript accordingly.

Reviewer 1 ·

Basic reporting

Satisfactory

Experimental design

Although the findings are in some ways not surprising and the veterinary profession has “stacked the deck” against itself, I wondered if in this article the authors contributed a potential bias too.
In the journals that were reviewed, one of the veterinary journals is clearly a research journal, i.e. American Journal of Veterinary Research. As veterinarians have access to research, laboratory, or university-owned animals that are not “real clinical patients”, I suspect it very unlikely that a clinical trial of real patients would have been submitted to this research journal. This journal had a large number of RCTs but unlikely to be of clinical patients. In contract the 5 medical journals are all primarily clinically focused as noted by the number of RCTs in Table 2.
As mentioned, I suspect the findings of the authors are still valid, but the numbers or methods do not seem without bias.

Validity of the findings

No comments

Additional comments

Specific comments:
Lines 118-120: What difference in proportions was used for this calculation? Specify.

Lines 188 and 189: Initials are used for an individual who contributed to the study but does not seem to appear as an author. This individual (affiliation and credentials) should be identified in some way. (I suspect it is the person mentioned in Acknowledgments, but it is important to indicate there was not a difference in the “quality” of those operators.)

Lines 216-217: These methods should be referenced. Do not use abbreviations, e.g. VIF.

Lines 225-228: Please clarify the reason for (justify) the large number of exclusions.

Line 240: The word phrase “less likely” followed by an OR of 3.8 (increased odds) seems contradictory. Reword this sentence for clarity.

Lines 246-249: These lines seem out of place and/or are a poor start for this paragraph/section. Suggest removing or placing in a more appropriate place in the Results.

Line 259: Parentheses should include “95% CI”.

Lines 264 and elsewhere: Although reporting of IQR is satisfactory, the reporting of a single number (difference) is not informative compared to reporting an actual range (xx – xx) so that positioning of data around the median is clearer. Please change to a range.

Line 267: A correlation coefficient (r) is not an appropriate expression of a “difference”; please use correct statistic.

Figure 2: I find this figure difficult to interpret, and this 3-D style is also known to distort perceived values. I do not find it of benefit and recommend deletion.

·

Basic reporting

This is a very well-written manuscript. In my review version of the manuscript some of the supplementary figures were missing labels that linked them to the text. Otherwise, the paper was easy to read.

The manuscript appears to be in a standard Peerj format.

Experimental design

The research question is meaningful and important. The investigation was rigorous and to a high technical standard. The methods, as described, are repeatable.

Questions:

1. Can the authors describe the "top journals" in a quantative way? Are the journals' impact factors high?

2. For the subgroup analysis, I'm not sure how many surgery papers are published in these journals. There may be lots of surgery RCTs published in surgery journals. So, is that a valid conclusion, about surgery RCTs?

Validity of the findings

My questions and comments about this manuscript are mainly about the underpinnings of the inference.

Inferential statistics use a random sample of data taken from a population to describe and make inferences about the population. For example, it is impractical to measure the diameter of every tree in a forest, but you can measure the diameters of a representative sample of trees and use that information to make generalizations about the diameters of trees in that forest.

In the study under review, It's not clear (to me) what is the representitive sample, and what is the population. What are all the articles found the search over a (2013) year a representitive sample of?

What population that I am making inference about?

It seems that the authors performed a census for the year 2013. If so, the P values aren't needed, are they?

Additional comments

Well done!

Are there any references that suggest that a poorly reported study is also a poor study? In other words, if a study fails to report its allocation methods, are the results of the study less accurate? Less valid? Less reliable? Are the study's inferences not right?

·

Basic reporting

The current article tackles an important issue in the veterinary profession today: The quality of publications which evaluate the effectiveness of interventions. However, in my view, the manner in which the authors have approached the issue has somewhat missed the significant points of the problem. Much effort is given to comparing the prevalence of RCTs among published EoI, while little effort was spent discussing the reasons why fewer veterinary RCTs were performed than medical RCTs (My position is discussed in more depth in the 'Validity' section below).

Additionally, the title is somewhat indelicate. I would recommend that it be changed to something along the lines of 'A comparison of the prevalence of RCTs among EoI between medical and veterinary literature'. The title should also reflect the restrictions of the study (e.g. articles were only taken from the year 2013, only 5 journals of each type were sampled).

Some specific points for edits are as follows:
* Lines 124-8: Provide the abbr used for these journals used later in paper.
* Line 133: The 10 journals included are described as 'Eligible journals', when these 10 were only the ones intentionally included. It would be more accurate to change this line to "the 10 selected journals".
* Line 226: No need for the serial semi-colons, remove and make separate sentences.
* Various places: places where the Chi-squared statistic should include the superscript 2, instead of merely 'Chi'.
* Table 1. Greater detail should be provided in the title.

Citations should be provided for the following statements:

* Citation for VIF and Condition index values (lines 215-7)
* Citation for Mann-Whitney U test (Line 218-9)

Experimental design

The sampling of papers for the study appears to be appropriate. The absence of blinding during the review is potentially concerning, but is addressed and would have made the research prohibitively difficult.

There are a few important statistical issues that need to be addressed.

* The power calculation was performed for two groups of equal sizes (Lines 182-7), but later the sampling methodology indicates the RCTs were sampled in a 1:2 ratio medical:veterinary.

*all of the veterinary RCTs were selected, so the journals were proportionally represented, but an equal number of RCTs were selected from each of the 5 medical journals (lines . It would be preferable to apply a single sampling methodology across both strata of journals to improve comparability. Any of the following options would be acceptable:

- take a set number of articles from all ten journals
- Weight each article by the # of EoI or RCT articles found the respective journal
- include all veterinary articles and sample RCTs from medical articles proportionately from each medical journal.

* In lines 210-7, it appears that confounders were assessed for inclusion based on statistical significance, but this method is inappropriate for assessing confounding. Covariates should be assessed for fit first using significance testing, but should also be assessed as possible confounders using A 'change of effect' method, instead of a hypothesis test. More information on assessing confounding in this matter is available in chapter 15 of Modern Epidemiology by Rothman, Greenland, & Lash or the article "Stratification by a multivariate confounder score" (Miettinen, Am J of Epi 1976).

* The selected journals were selected to be representative of the literature as a whole, but were considered to be independent and were assessed using fixed effects (lines 210-2). The manner in which the articles within each journal are likely similar to each other, and hence the articles nested within each journal cannot really be assumed to be independent of each other. The of effect of the 'journal' factor should really be included as a random effect. Additionally, since the power calculations provided assume the sampled elements are independent and the assumption of independence is not likely met, the provided power calculations are likely invalid.

* Non-parametric methods were used to compare the number of patients in studies, while parametric analytic method were used elsewhere. Justification should be provided. Even if the observations aren't normally distributed, the central limit theorem indicates that the mean is normally distributed. If there is interest in comparing the group medians, other justification should be provided.

* The purpose of comparing the prevalence of EoI articles between the veterinary and medical literature is unclear (lines 239 - 42) since non-EoI are not within the scope of this paper. The relative proportions may be retained in the manuscript, but the hypothesis testing between the two groups should be removed.

Validity of the findings

As mentioned above, the comparison of the prevalence of RCTs among EoI studies between human and veterinary literature may not be a valid. While this manuscript presents compelling evidence that the human medicine literature does tend to publish proportionally more RCTs per EoI study, I don't find the comparison to be particularly meaningful, nor do I see that compelling evidence has been presented for such comparison to be made. The reason I would question the validity of the comparison is due a number of inherent difference between the medical and veterinary professions.

Foremost, RCTs are expensive, the amount of funding available to human medical research will *always* outpace the funding available for veterinary research. While it cannot be denied that RCTs provide results with greater external validity than observational studies, the results cost-benefit of observational studies cannot be denied. Since the latter are more cost effective and veterinary funding is limited, veterinary research will *always* perform proportionally fewer RCTs, and hence will *always* be 'less valid' than medical literature by this measure.

The variety of species in veterinary medicine additionally makes performing RCTs on every species cost prohibitive. Even if we restrict the discussion to small animal medicine (which, of all divisions of veterinary medicine, probably most closely resembles human medicine), RCTs for cats and dogs would have to be performed separately, doubling the number of studies that need to be performed to maintain an 'equal' level of validity with human medicine.

Second, the risk of less valid results (the cost when they are made) is greater in human medicine than in veterinary medicine. Even speaking as a veterinarian, I must concede that the value of a human health exceeds the value of animal health, hence it is more acceptable to have, on average, somewhat less valid studies in veterinary medicine than than in human medicine. This is reflected in the various guidelines that exist for the publication of RCTs in human medical journals, but are less common (or do not exist) in veterinary publication guidelines.

The goals in veterinary medicine are also substantially different than in human medicine, e.g. in production medicine, productivity is the primary metric of treatment success. The cost of lost productivity due to less valid study results is of less importance than the loss of life or quality of life due to less valid study results in human medicine.

As a related note, I am a bit concerned about the omission of the *type* of veterinary medicine (e.g. small animal, equine, ruminant, etc) in the manuscript; the disparate goals of the various veterinary disciplines would almost certainly lead to heterogeneity in the proportions of RCTs per EoI study, meaning an estimate pooled across all disciplines wouldn't necessarily be valid (i.e. the pooled estimate may not be representative of veterinary literature in total).

In summary, the validity of comparing the proportion of published EoI studies that are RTCs between veterinary and human literature may not be a valid comparison since the need for a higher quality of results in human medicine is greater than the same need in veterinary medicine. At the very least, the manuscript should include a discussion covering the points above to explore possible reasons that the differences exists between the human and veterinary literature.

I believe a more important metric for comparing the validity of studies across the human-veterinary split is how the veterinary literature reports RCTs compared to how they are reported in human studies. Examples include reporting factors such as a priori power calculations, randomization schemes, allocation concealment, etc. These are important factors that need to be reported better in veterinary literature, and manuscript highlights these deficiencies well.

Other thoughts on validity:

* The authors spent a fair bit of text in the introduction discussing how reporting of EoI/RCTs changed over time in the human literature, but little time was spent discussing similar trends in the veterinary literature. This would be useful to further contrast the state of veterinary literature to medical. Perhaps the basis of a follow up paper.

Additional comments

Overall, the topic covered by the paper is timely and important, however I believe refocusing the paper will be of great benefit to both the manuscript and the veterinary profession. More attention should be spent on explaining the reasons that the differences between medical and veterinary medicine exist.

---

## Round 0.2 · Minor Revisions

Thank you for responding to the reviewers and their comments. Although the manuscript is improved, there are a number of remaining issues that I am requesting you address. The first three are my own:

First, the manuscript has numerous grammatical errors through it (e.g., “On the basis of our selection criteria, it was included ‘AJVR’… .”). I recommend that you have an independent individual review your manuscript for grammatical errors prior to re-submission.”

Second, although it is not necessary to provide chi-square test statistics, if you choose to do so you are obligated to accompany them with the tests’ degrees of freedom; otherwise they have no interpretability.

Third, I found the distinction between “veterinary” and “medical” disciplines confusing. Please alter these to “veterinary medical” and “human medical” disciplines to avoid confusion to readers.

The following are comments from the reviewers that need additional explanation or elaboration:

Reviewer 1, comment 2: Discussion needs elaboration about limitation of using only these 10 journals, and not just what the effect of including AJVR is.

Reviewer 1, comment 4: Just remove the initials. Names can remain in the acknowledgments.

Reviewer 1, comment 7: However, specifically explain why the exclusions were made in the following statement on Lines 249+: “Finally, we included only “explicit RCTs”, i.e., randomized trials, self-defining RCTs or trials registered in a repository and we excluded all the cross-over and cluster RCTs.” Also, define “self-defining RCT”, “cluster RCTs”. A cross-over trial can still be randomized, so a justification is necessary.

Reviewer 1, comment 13: I found the graph to be somewhat confusing, though not fatally so. On the other hand, the explanation of the graph contributes to misunderstanding. Here is what is written: “Association (Odds ratio and 95% CI) between randomization and type of intervention (surgical/non-surgical). Surgical interventions were more likely to be non-randomized in both disciplines.” However, as an example, the odds ratio of 3.7 is not between randomization and type of intervention: it appears to be between “discipline” and type of intervention. In addition, the discipline term “medical” is undefined and requires elaboration.

Reviewer 2, comment 17: On Line 140, please indicate the range of impact factors for the five journals in the veterinary and “medical” journal categories.

Reviewer 2, comment 18: Please note in the Discussion the limitation that an unknown number of RCTs may be published in in surgery journals, and how that may affect the interpretation of this research.

Reviewer 2, comment 19: Please include information from your rebuttal in your Discussion.

Reviewer 3, comment 24: I agree with the reviewer on the first part and with you on the second part. Your null hypothesis is that there are no differences in effectiveness between veterinary and human medical studies. Rejection of the null does not prove the alternative hypothesis. I offer the following as a compromise title: “Effectiveness of intervention studies in the human and veterinary medical literature: a cross-sectional survey of ten leading veterinary and human medical journals '

---

## Round 0.3 · Minor Revisions

The remaining comments I have are largely about improving the language of the manuscript in certain places to enhance reader understanding. All other comments have been satisfactorily addressed, for which I thank the authors.

Line 110: Please change "Furthermore, opinion leaders have recently underlined that animal trials should be more similar to human RCTs (Muhlhausler et al., 2013)." to "Furthermore, opinion leaders have recently underscored that animal trials should be more similar to human RCTs (Muhlhausler et al., 2013)."

Line 249: Remove the apostrophes from Article.

Line 283: Please change "Sensitivity analyses were conducted excluding from the analysis certain types of randomized trials in order to determine their effect on the final results." to "Sensitivity analyses were conducted by excluding from the analysis certain types of randomized trials, in order to determine their effect on the final results."

Line 286: It is unclear what you mean by "poor reporting". Are you writing that the methods used for randomization are poorly described? The same question applies to line 288 (i.e., "poor methodological reporting").

Line 292: It is unclear what you mean by "attention."

Line 292: Please change "In the final analyses, we excluded all the cross-over RCTs and cluster RCTs because also these trial designs have been associated with poor reporting (Walleser et al., 2011; Straube et al., 2015)." to "In the final analyses, we excluded all the cross-over RCTs and cluster RCTs because these trial designs have also been associated with poor reporting (Walleser et al., 2011; Straube et al., 2015)."

Line 390: Is 0.87 a P-value? If so - please indicate this as P=087.

Line 391: Please provide a reference to Nagelkerke's r-squared, as readers may not be familiar with this. Also, the abbreviation AUC is undefined - please note that this is area under the curve.

Line 506: The sentence "This is not unexpected, as surgical RCTs pose more than pharmaceutical RCTs (McCulloch 2009)." is not grammatically correct; there appears to be a word missing.

Line 509: Please change "However, it is possible that surgical teams aim to submit the results of surgical RCTs to specialist surgical journals instead that to broad scope journals." to "However, it is possible that surgical teams prefer to submit the results of surgical RCTs to specialist surgical journals instead of to broad scope journals."

Line 536: By "n. 133" do you mean "n=133"? The same question holds for "n. 154".

Figures and Tables: Please define all abbreviations that are being used (including journal names, Eol, M-H, OR, CI, etc.). Also, it is not necessary to capitalize odds ratio i the header to Figure 2. These figures and tables should be completely understandable without having to refer to the text in the body of the manuscript.

---

## Round 0.4 · accepted · Accept

Thank you for making your changes so quickly. All my comments have now been addressed and I am happy to recommend acceptance of your manuscript.